# Learning to Cooperate and Communicate Over Imperfect Channels

## Abstract

Information exchange in multi-agent systems improves the cooperation among agents, especially in partially observable settings. This can be seen as part of the problem in which the agents learn how to communicate and to solve a shared task simultaneously. In the real world, communication is often carried out over imperfect channels and this requires the agents to deal with uncertainty due to potential information loss. In this paper, we consider a cooperative multi-agent system where the agents act and exchange information in a decentralized manner using a limited and unreliable channel. To cope with such channel constraints, we propose a novel communication approach based on independent Q-learning. Our method allows agents to dynamically *adapt* how much information to share by sending messages of *different size*, depending on their local observations and the channel properties. In addition to this message size selection, agents learn to encode and decode messages to improve their policies. We show that our approach outperforms approaches without adaptive capabilities and discuss its limitations in different environments.

## 1 Introduction

In multi-agent systems, cooperation and communication are closely related. Whenever a task requires agents with partial views to cooperate, the exchange of information about one's view and intent can help to reduce uncertainty and allows for more well-founded decisions. Communication allows agents to solve tasks more efficiently, and can even be necessary to achieve acceptable results (Singh et al., 2019). As an example, consider a safety-critical autonomous driving scenario (Li et al., 2021). By letting the cars exchange sensor data or abstract details about detected objects in the scene, occluded objects can be considered in the planning processes of all cars and reduce the risk of collisions.

Multi-agent reinforcement learning (MARL) comprises learning methods for problems where multiple agents interact with a shared environment (Buşoniu et al., 2010; Hernandez-Leal et al., 2019). The goal is to find an optimal policy for the agents that maximizes the outcome of their actions with respect to the environment's reward signal. Key challenges in MARL include non-stationarity (Papoudakis et al., 2019), the credit assignment problem (Zhou et al., 2020) and partial observability (Oroojlooyjadid & Hajinezhad, 2019). We focus on cooperative environments with partial observability.

As communication is essential in cooperative environments, many works include a predefined information exchange between agents (Melo et al., 2011; Schneider et al., 2021). Additionally, there is ongoing research to include learnable communication into MARL approaches. Pioneering work gave first empirical evidence that communication between agents can be learned with deep MARL (Foerster et al., 2016; Lowe et al., 2017; Sukhbaatar et al., 2016). This enhances the performance on existing environments and allows to address a new class of problems that require communication between agents. Building upon these ideas, many researchers proposed methods to improve the performance and stability of these approaches (Gupta et al., 2020; Jiang & Lu, 2018; Li et al., 2021). While related work investigates effects of using different fixed message sizes (Li et al., 2022) and multiple communication rounds (Das et al., 2019), selectively sending messages (Singh et al., 2019), and sending messages only to other agents in their proximity (Jiang & Lu, 2018), most of these approaches are designed for communication channels without capacity limitations or message losses. Recent approaches started to investigate such settings, e.g. by learning central controllers for coordinated

access to a communication channel (Kim et al., 2019). To the best of our knowledge, there are no studies on message size adaptation to improve multi-agent communication over imperfect channels.

In our work, we address this gap by investigating a cooperative MARL setting in which agents communicate over an unreliable and limited channel. We focus on agents learning when, what and how much to communicate over imperfect channels in a decentralized manner. The key challenge here is to determine how to utilize the limited capacity efficiently and cope with the lack of reliability, in order to maximize the benefit for the cooperative multi-agent problem.

With this paper, we provide a novel approach to address this challenge. Our contributions are as follows: (i) we propose a novel communication approach that allows for an adaptive message size selection while learning the message encoders and decoders, (ii) we introduce discrete communication trained with the pseudo-gradient method, (iii) we analyze the effect of different message types and message sizes, (iv) we introduce POMNIST, a fast MNIST-based benchmark environment for communication, (v) we show that agents adapt to the given communication channels in POMNIST and show limitations of our approach in the traffic junction environment.

## 2  RELATED WORK

Agents in MARL can exchange information a) with implicit communication, and b) with explicit messages that are forwarded between the agents. Implicit communication refers to exchange of information without separate communication actions, e.g. through the agents' regular actions and observations (Foerster et al., 2019) or with a joint policy (Berner et al., 2019).

Within the scope of this paper, we focus on explicit communication. This can further be divided into i) continuous communication with real-valued messages and ii) discrete communication with a finite set of messages. In the context of deep learning, exchanging continuous messages allows for back-propagation across different agents (Sukhbaatar et al., 2016). This results in significant performance improvements in partially observable environments, where agents can benefit from coordination or the exchange of local information. Recent approaches include restricting communication to agent groups (Jiang & Lu, 2018) and specific topologies (Du et al., 2021), deciding when to send messages (Liu et al., 2020; Singh et al., 2019) and estimating the importance of messages with attention (Das et al., 2019; Li et al., 2021; Rangwala & Williams, 2020).

Discrete communication with finite message sets allows for more fine-grained control of the used data rate in limited communication scenarios and is the focus of this paper. In order to facilitate backpropagation for discrete communication, Foerster et al. (2016) regularize continuous messages with noise during training and discretize them during evaluation. In their experiments, this yields better results than extending the action space with communication actions. Differentiability can also be retained by sampling messages from a gumbel-softmax distribution (Jang et al., 2017) instead of a categorical distribution (Gupta et al., 2020; Lowe et al., 2017). Li et al. (2022) aim to compensate for the message discretization with skip connections. Their results also suggest that the message size has a neglectable effect on continuous and a significant effect on discrete communication.

Our work combines learnable communication with deep Q-learning to adaptively select the message size based on the observations given at each step. We consider an uncoordinated channel of limited capacity and demonstrate how agents can benefit from message size selection in such settings. Related works consider limited communication via a centrally controlled channel of limited capacity (Kim et al., 2019), by pruning messages (Mao et al., 2020) or with regularizations based on the length of messages (Freed et al., 2020). Hu et al. (2022) show empirically that controlling whether to send messages improves communication in a slotted $p$-CSMA channel. It is unclear how and if agents can choose from different message sizes to improve their communication efficiency in unreliable channels of limited capacity. We address this research gap with our work.

Although not the focus of this work, efficient use of imperfect channels can also be improved by coordinating the agents' access to the channel. For example, Kim et al. (2019) and Wang et al. (2020) consider this by learning a centralized scheduler for multi-agent communication. The classical communication literature compromises a multitude of sophisticated mechanisms for medium access control (Huang et al., 2013; Kumar et al., 2018). We note that such schemes can be used in conjunction with our adaptive message size selection and leave further exploration of this combination to future work.

## 3 ADAPTIVE COMMUNICATION

In this section, we formulate the cooperative multi-agent problem with message size selection and describe our adaptive communication approach. Fig. 1 summarizes the information flow in the multi-agent system with adaptive communication.

### 3.1 PROBLEM FORMULATION

We consider a network of $N$ agents in a cooperative and partially observable stochastic game (Hansen et al., 2004). Each agent $i \in I := \{1, ..., N\}$ has a private and partial observation $o_t^i \sim \mathcal{O}^i(o_t^i \mid s_t)$ at time step $t$ based on the state $s_t \in \mathcal{S}$. As none of the agents has direct access to the state, it is essential to communicate with others in order to make better decisions. However, the agents are not provided with any prior information on how to communicate. During training, they must learn to exchange meaningful and beneficial information and correctly decode the incoming messages.

We consider stepwise communication where each agent sends a single message $m_t^i \in \mathcal{M}^\varphi$ of size $\varphi \in \Phi \subseteq \mathbb{N}_0$ to all agents. Note that a message of size zero corresponds to not sending any message. Each agent can choose a different size $\varphi_t^i$ in each step, but we omit superscript and subscripts in this section for notational simplicity. The message space $\mathcal{M}$ can be continuous $\mathcal{M} = [-1, 1]$ or discrete $\mathcal{M} = \{0, 1\}$. Once all agents have created a message, the messages are transferred using a given channel. The success of sending a message depends on the channel model, which receives all messages at the current timestep as input and outputs the set of successfully transmitted messages $M_t \subseteq \{m_t^1, ..., m_t^N\}$. In each step, agents receive messages that have been successfully transmitted in the previous step, except for their own message $M_{t-1}^i := M_{t-1} \setminus \{m_{t-1}^i\}$. Each agent takes an action $a^i \in \mathcal{A}$ sampled from its stochastic *action policy* $\pi^i$ based on its observation $o_t^i$ and incoming messages, $a_t^i \sim \pi^i(a_t^i \mid o_t^i, M_{t-1}^i)$. The actions are then applied in the environment which causes it to transition to the next state based on the transition probabilities, $s_{t+1} \sim \mathcal{P}(s_{t+1} \mid s_t, \boldsymbol{a}_t)$ with $\boldsymbol{a}_t := (a_t^1, ..., a_t^N)$. In parallel to taking an action, each agent chooses a message size $\varphi$ from the given set of sizes $\Phi$ using its stochastic *message size policy* $\pi_\Phi^i$ conditioned on the current observations and incoming messages, $\varphi \sim \pi_\Phi^i(\varphi \mid o_t^i, M_{t-1}^i)$. The agent then creates a message $m_t^i \in \mathcal{M}^\varphi$ of size $\varphi$ with a size-specific message encoder $m_t^i = f_\varphi^i(o_t^i, M_{t-1}^i)$. The messages are broadcasted to all other agents using the given communication channel and have no direct influence on the environment.

Each agent receives individual rewards $R^i(s_t, \boldsymbol{a}_t) \in \mathbb{R}$ depending on the state of the environment $s_t$ and the joint action $\boldsymbol{a}_t$. The discounted return of agent $i$ for step $t$ is defined as $G_t^i = \sum_{k=t}^T \gamma^{k-t} R^i(s_k, \boldsymbol{a}_k)$ with a time horizon $T \in \mathbb{N}$ and discount factor $\gamma \in [0, 1]$. We consider cooperative environments in this paper. Agents cooperate by choosing actions that increase the rewards of other agents and by exchanging local information via messages. Each agent $i$ optimizes their action policy to maximize their return $G_0^i$. This is complemented with finding message size policies $\pi_\Phi = \{\pi_\Phi^1, ..., \pi_\Phi^N\}$ and message encoders $f_\Phi = \bigcup_{i \in I, \varphi \in \Phi} \{f_\varphi^i\}$ that jointly maximize the expected mean discounted return of all agents for a given channel, i.e. $\max_{\pi, \pi_\Phi, f_\Phi} \mathbb{E}\left[\frac{1}{N} \sum_{i=1}^N G_0^i\right]$.

### 3.2 ARCHITECTURE

With adaptive communication (see Fig. 1), we propose a deep MARL architecture to jointly learn action policies, message size policies and message encoders, only based on the reward signal. Our approach comprises two modules to learn communication: (a) a message-size selector and (b) a message encoder. This is complemented with an action selector to perform actions in the environment. We first describe how the agent's input is processed and then explain the modules.

**Input processing** The input consists of the observation from the environment $o_t^i$ and the successfully transmitted messages from the last step, excluding the agent's own message $M_{t-1}^i$. An environment-specific observation decoder maps the observation to a vector of fixed length. The message decoder maps all incoming messages to a vector of fixed length by taking the mean of all messages padded to the same length, including a one-hot encoding of the message's length. The vectors from the observation and message decoders are concatenated and processed by a core module. Details regarding the concrete architecture can be found in the appendix in Sec. A.1. The core's output $x_t^i$ is the input to the following modules.

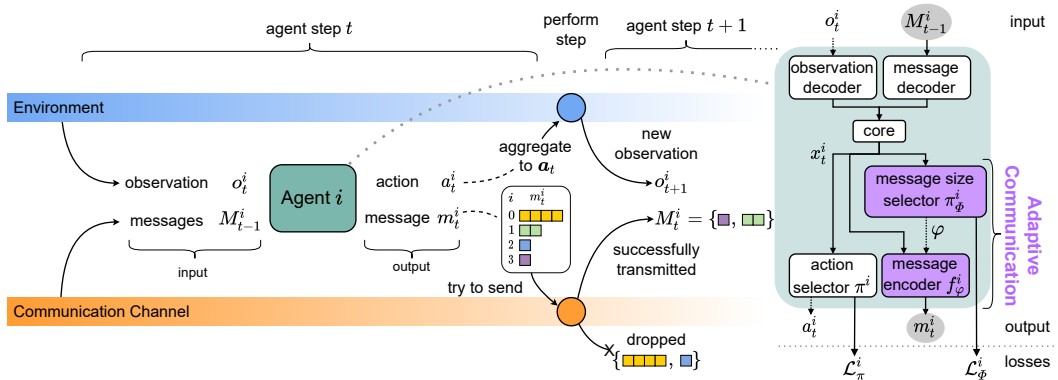

Figure 1: Overview of the multi-agent system with adaptive communication. At each step, agents receive an observation and a set of messages. Based on this input, they select an action and create a message of selected size $\varphi \in \Phi$. Actions are used to perform steps in the environment, messages are distributed stepwise according to a given channel model. The channel defines whether messages are dropped or transmitted correctly. The gradients from the action and message size selector losses of an agent that receives a message are backpropagated to the sender of this message.

**Message size selector**  The message size selector employs independent deep Q-networks (Tampuu et al., 2017), where each agent $i$ independently and simultaneously learns its own Q-function. In our case, we want to select the message size that leads to the highest expected discounted returns for the receiving agents. The message-size-value network $Q^i(x_t^i, \varphi; \theta_\Phi)$ conditions on the agent's individual core output $x_t^i$ and returns a value for each message size $\varphi \in \Phi$. It is parameterized by $\theta_\Phi$.

The message size selection cannot have any effect on the rewards of the current step, as the created messages will be received in the next step. Therefore, we propose to use an offset of one step in the target for the message-size values. Additionally, the sending agent does not receive its own message in the next step, the sent message cannot have any effect on this reward either. Based on these considerations, we define the message-value target of agent $i$ in step $t$ for message size $\varphi$ as

$$D_t^i(\varphi) := \mathbb{E}\left[\frac{1}{N}\left(\sum_{j=1}^{N} G_{t+1}^j - R^i(s_{t+1}, \boldsymbol{a}_{t+1})\right) \;\middle|\; \varphi_t^i = \varphi\right]. \tag{1}$$

We train the message-size-value network to approximate the target in Eq. 1 by iteratively minimizing the mean squared error over multiple sampled training episodes for all agents, see Eq. 2.

$$\mathcal{L}_\Phi^i(\theta_\Phi) := \mathbb{E}\left[\left(Q^i(x_t^i, \varphi_t^i; \theta_\Phi) - D_t^i(\varphi_t^i)\right)^2\right] \tag{2}$$

The agent's message selection policy $\pi_\Phi^i$ uses Boltzmann softmax exploration with decreasing temperature $\eta \to 0$ (Sutton & Barto, 2018) based on the predicted message-size values, i.e. message size $\varphi \in \Phi$ is chosen with a probability proportional to $\exp(Q^i(x_t^i, \varphi; \theta_\Phi)/\eta)$.

**Message encoder**  After the message size $\varphi$ is selected, the agent creates its message $m_t^i$ using a *message encoder* network $m_t^i = f_\varphi^i(o_t^i, m_{t-1}; \xi_\varphi)$ parameterized by $\xi_\varphi$. For a given set of available message sizes $\Phi$, the agent learns to encode messages in each message size $\varphi \in \Phi \setminus \{0\}$. This is achieved through a multi-headed neural network instead of separate networks, which allows learning both shared and private parameters for different message sizes while reducing the complexity of the model. It should be noted that the agents learn how to form messages for all message sizes during training, which is the key idea leading to adaptivity in this work. We describe the *message types* that define the output of the message encoder in Sec. 3.3.

**Action selector** The action selector represents the agent's policy $\pi^i$ and is jointly learned with message selection and encoding, but it is conceptually independent and not the focus of this paper. We evaluate our approach with (a) an $\epsilon$-greedy action selector based on Q-values that are trained with the discounted monte carlo return $G_t^i$ as target and (b) a stochastic action selector similar to (Singh et al., 2019) that is trained via REINFORCE (Williams, 1992), including a learned value baseline and an entropy bonus loss to encourage exploration. Each action selector has its own loss function $\mathcal{L}_\pi^i$ to maximize the agent's individual return. This loss is backpropagated through the core to train the observation and message decoders. Through the message decoder, the gradients are passed to the messages' senders if supported by the message type. This enables the sending agents to adapt the content of their messages to improve the actions of the recipients.

**Parameter sharing** We employ centralized learning and decentralized execution (Foerster et al., 2016; Kraemer & Banerjee, 2016; Oliehoek et al., 2008; Rashid et al., 2018) to minimize the mean losses $L_\Phi^i$ and $L_\pi^i$ over all agents. During learning, the trainer has access to the reward of all agents to optimize the message-size policies. These policies are represented by a single model. However, the execution is performed in a fully decentralized manner. The agents make the decisions for actions and message size solely based on their individual observations and the incoming messages. We allow agents to learn different behavior by adding agent-specific features to the observations. This approach reduces the number of parameters and is computationally more efficient.

## 3.3 MESSAGE TYPES

Messages are either continuous or discrete and the choice of encoding method can affect the communication. We refer to the combination of message space and encoding method as a *message type*. In this section, we present the message types that we consider in conjunction with our approach.

**Discrete communication with Q-values** Communication can be seen as an extension to the agent's action space. Accordingly, standard RL methods like Q-learning can be applied to learn communication actions (Foerster et al., 2016). We use the mean discounted return of the receiving agents equivalent to our message size selection to learn Q-values for each message value. Therefore, this message type is not differentiable and the agents have to learn how to encode messages purely by trial and error. To avoid confusion, the loss for this message type is omitted in Fig. 1. This method does not scale well, as increasing the message size leads to an exponential increase in the number of messages values and message Q-values that have to be learned.

**Continuous communication** With this message type, agents send real-valued messages to each other. As they are used as input to networks representing different agents, real-valued messages allows gradient flow across agents (Foerster et al., 2016), as indicated on the right side of Fig. 1. This serves as direct feedback on messages from the recipients to the sender and allows the message encoder network to learn from this feedback. Since the real-valued messages are the most expressive message type used in this work, we expect it to achieve the best results and to serve as baseline.

**Discrete communication with pseudo-gradient method** The exchange of discrete messages with Q-values limits the agents, since the discrete communication channel is non-differentiable and the backpropagation algorithm cannot be applied for end-to-end training across agents. The pseudo-gradient method was first proposed for recurrent neural networks with discrete activations (Zeng et al., 1993) and later applied to multi-layer neural networks (Goodman & Zeng, 1994). This method approximates the gradient of the discrete activation function using the true gradient of an analog activation function as a heuristic hint. In our work, we employ the *pseudo-gradient* method with $PG(m) := 2 \cdot \mathbb{1}\{\tanh(m) > 0\} - 1$ for message features $m$ in the forward pass and directly pass the gradients to $\tanh$ in the backward pass. This allows to benefit from end-to-end backpropagation.

**Discrete communication with DRU** The discretise/regularise unit (DRU) proposed by Foerster et al. (Foerster et al., 2016) aims at retaining differentiability with discrete messages. During training, the DRU regularizes continuous messages $m$ by adding noise $\text{DRU}_{\text{train}}(m) = \text{Logistic}(\mathcal{N}(m, \sigma^2))$ from a normal distribution $\mathcal{N}(m, \sigma^2)$ with mean $m$ and standard deviation $\sigma$. The authors argue that noise with $\sigma > 2$ effectively regularizes the message to a single bit. During execution, the DRU then discretizes the message using an element-wise threshold $\text{DRU}_{\text{execution}}(m) = \mathbb{1}\{m > 0\}$.

### 3.4 LIMITED COMMUNICATION CHANNEL

Communication resources are limited in practice. We propose to combine the problem of learning how to communicate with a limited and lossy communication channel. While greater message sizes typically yield better performance in an unlimited scenario, always selecting the highest message size might lead to congestion and dropped packets in real networks. Missing and delayed messages could have a negative impact on the cooperation. Instead of assuming a perfect communication channel like most related work, we consider a limited channel with stochastic collisions between messages.

We use a slotted channel model that is parameterized by a channel size $C \in \mathbb{N}$ defining the available slots $\{0, ..., C-1\}$. The communication is synchronized with the environment's steps. If an agent decides to send a message of size $\varphi \in \Phi \setminus \{0\}$, this message is inserted into $\varphi$ contiguous slots using a fixed stochastic channel access mechanism. Messages are inserted into the channel independently and the starting slot is chosen uniformly from $\{0, \varphi, 2\varphi, \ldots, \lfloor \frac{C-\varphi}{\varphi} \rfloor \varphi\}$. Messages that do not fit into the communication channel, i.e. $\varphi > C$, are dropped. After the insertion of all messages, if two or more messages shares at least one slot, a collision occurs and all involved messages are dropped.

We have chosen this slot assignment over a completely uniform assignment to reduce the number of collisions in the channel. However, note that the channel is still highly unreliable and the expected throughput is way lower than the channel size, depending on the agent's message sizes. This is a known phenomenon of early decentralized slotted channel access models with uncoordinated clients (Tanenbaum & Wetherall, 2011). More details are provided in the appendix in Sec. A.5.

This simple channel model suffices to study the behavior of learning agents with limited and unreliable communication, as we will show in the experiments section. More sophisticated channels and channel access mechanisms could be considered in future work.

## 4 EXPERIMENTS

In this section, we discuss the effectiveness of adaptive communication in different environments. We conduct all experiments with two Intel® Xeon® Silver 4214 and an NVIDIA® GeForce® RTX 2080 Ti.

### 4.1 PARTIALLY OBSERVABLE MNIST (POMNIST)

We first evaluate our approach with cooperative digit prediction based on the MNIST dataset (LeCun et al., 1998). It consists of grayscale images of hand-written digits with $28 \times 28$ pixels and their corresponding labels. It is split into train and test sets with $60{,}000$ and $10{,}000$ examples respectively.

**Environment**   We introduce the partially observable MNIST (POMNIST) environment that splits each MNIST sample into partial views of the same size (see Figure 2). These views are fixed during training and evaluation, and each agent is assigned to one particular view. Each episode consists of two steps: In the first step, agents observe their local view and broadcast a message to all other agents. In the second step, the agents independently predict the digit based on their observation and the received messages. Agents receive a reward of 1 if their prediction is correct and -1 otherwise. The agent's reward in the first step is set to zero. The mean return can be mapped to accuracy with *accuracy = (mean return + 1)/2*.

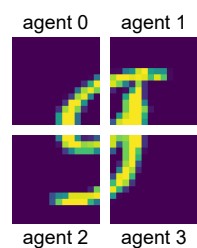

Figure 2: A digit is split into non-overlapping views for four agents.

POMNIST allows to construct environments of varying difficulty and reliance on communication. The higher the number of views and agents, the less information is included in each agent's observation. Therefore, they have to increasingly rely on communication to make correct predictions.

**Training**   We use the Q-value action selector and train our approach for 2000 iterations using the Adam optimizer (Kingma & Ba, 2015) and learning rate 0.001. The core is a dense layer with a skip connection. We train with 2048 parallel environments, resulting in a batch size of 2048 for each agent in each iteration. During training, we randomly draw samples with replacement from the train set. For testing, we run exactly one episode for each sample in the test set. A complete training run takes around 2 minutes without communication to 4 minutes with adaptive communication. The message-size selection begins with an exploration phase until iteration 1200, details are in Sec. A.1.

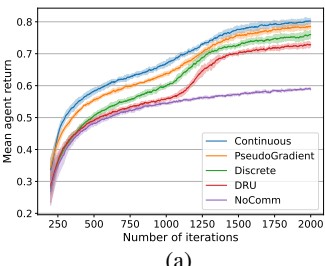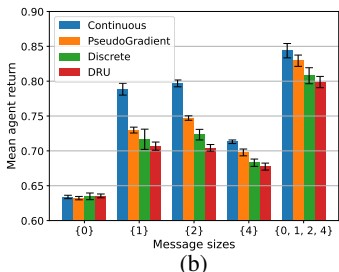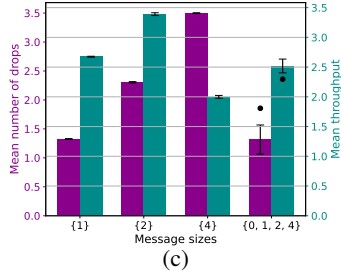

Figure 3: Results for training (a) and testing (b, c) in POMNIST with 4 agents and channel size 8. (a) shows mean return during training, (b) is mean return of different message types with fixed and adaptive message sizes, and (c) shows channel metrics for pseudo-gradient messages. Each line and bar shows the mean over 5 runs, the transparent area and the whiskers represent the standard deviation. The black dots in (c) show the metrics' values for a random message size selection.

**Results** A single agent with full view achieves a mean return of $0.979 \pm 0.002$ on the test set. We consider a configuration with one horizontal and one vertical split, resulting in 4 agents. The mean return on the test set for 4 agents without communication is $0.633 \pm 0.003$. We focus on message sizes $\Phi \subseteq \{0, 1, 2, 4\}$ and explain this selection in the appendix, see Sec. A.2.

To investigate how adaptive communication performs in limited channels and how it compares to fixed message sizes, we consider a case example with a channel of size 8. Fig. 3a shows the mean return of our adaptive approach with message sizes $\Phi = \{0, 1, 2, 4\}$ during training for different message types. Confirming our initial expectations, the continuous messages show the best performance, followed by discrete communication with the pseudo-gradient method. The DRU shows the lowest performance.

Fig. 3 (b) and (c) show the results on the test set after training. Fig. 3b shows that adaptive communication achieves higher returns than the approaches with fixed message sizes. For the analysis of the channel metrics in Fig. 3c, we focus on the pseudo-gradient message type as it achieved the highest return among the discrete communication methods. The left bars show the mean number of dropped messages in each step and the right bars show the throughput. The throughput is defined as the average number of slots occupied by messages that are not colliding. The number of dropped messages increases proportionally to the message size for single message sizes. The throughput increases from message size 1 to 2, but then decreases from 2 to 4 due to the high number of collisions in the channel. It can be seen that the number of drops and the throughput for adaptive communication are comparable with the results for message size 1. When compared to a random selection of message sizes in $\{0, 1, 2, 4\}$, shown as black dots, we can see that adaptive communication decreases the number of drops on average and increases the throughput. It is noticeable that the standard deviations for the mean number of drops and throughput in Fig. 3c are considerably higher for adaptive communication. This can be seen as an indication that individual runs might learn different strategies for the message size selection, while still outperforming the approaches with single message sizes.

Next, we analyze the effect of different channel sizes by experiments with adaptive message sizes $\Phi = \{0, 1, 2, 4\}$ using the pseudo-gradient method. Fig. 4 visualizes the throughput during training and Tab. 1 shows the performance metrics on the test set. In all cases, the agents are able to increase the throughput with adaptive communication, compared to the exploration phase. When using an unlimited channel, the throughput quickly reaches the highest value after the initial exploration phase. As expected, this leads to the highest return. By introducing the limited communication channel, the probability of a successful transmission decreases with the channel size. This can be observed both from the increasing number of dropped messages and from the decreasing throughput. The limited communication ability is also reflected in the decreased mean return, proportional to the channel size.

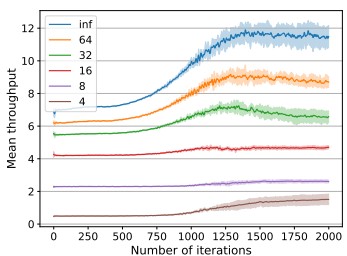

Figure 4: Mean throughput during training with different channel sizes. Each line shows the mean over 5 runs, the transparent area represents the standard deviation.

Table 1: Performance metrics for pseudo-gradient messages with different channel sizes over 5 runs.

| Channel size $C$ | inf | 32 | 8 | 4 |
|---|---|---|---|---|
| mean return | **0.94** ±0.00 | 0.88 ±0.00 | 0.83 ±0.01 | 0.80 ±0.03 |
| # of drops | 0.00 ±0.00 | 0.63 ±0.05 | 1.23 ±0.23 | **1.36** ±0.42 |
| pos listening | **0.26** ±0.01 | 0.17 ±0.01 | 0.14 ±0.00 | 0.13 ±0.03 |
| pos signaling | 0.66 ±0.05 | 0.75 ±0.02 | **0.83** ±0.03 | 0.69 ±0.07 |
| throughput | **11.84** ±1.12 | 6.55 ±0.35 | 2.60 ±0.10 | 1.55 ±0.29 |
| mean msg size | **2.68** ±0.31 | 2.16 ±0.14 | 1.62 ±0.29 | 1.39 ±0.15 |

The *positive listening* metric (see Sec. A.3) quantifies the impact of communication on an agent's behavior. As expected, this impact is highest for the unlimited channel, since the agents send messages of higher sizes more frequently and are able to share more information. The *positive signaling* metric (see Sec. A.4) quantifies the correlation between an agent's outgoing messages and its actions. Interestingly, positive signaling increases with decreasing channel size, except for channel size 4. As the agents send fewer messages as the channel size decreases, this could lead to them sending messages that are more specific to their local information. However, upon inspection of our data, we see that the agents restrict themselves to sending messages of size 1 and 2 with channel size 4. These message sizes might not be enough to encode specific situations, which could explain the decrease in the positive signaling for channel size 4. We observe that even in an unlimited channel, the mean throughput is below 12 instead of its maximum value of 16. The mean message size in Tab. 1 shows that the agents do not always send messages of the maximum size 4. This could indicate that agents learn to judge the relevance of their local information, and that they avoid confusing other agents by sending shorter messages when this local information cannot contribute to the success of the team. We also hypothesize that the agents encode information in the selected message sizes. The following ablations support this such that the agents perform better with adaptive message sizes and zero content compared to randomly selected sizes.

We show an ablation study of our approach in Fig. 5. The bar *none* is the baseline with message size $\Phi = \{0\}$. The others are adaptive communication with message sizes $\Phi = \{0, 1, 2, 4\}$, $C = 8$ and the following modifications: *random* selects random message sizes but learns the message encoders, *zeros* learns the message size selection but the encoders always return zero, *adaptive* is our approach that jointly learns the message size selection and message encoder. The increased return from none to random shows the benefit of learning message encoders and the increment from random to adaptive shows the benefit of the adaptive message size selection. Surprisingly, sending messages with content zero of different sizes outperforms the random mode. This indicates that the message size itself can be used to share information between the agents. As there are four different message sizes, selecting a message size can be seen as discrete communication with 2 bits of information. However, the information about a message's size is lost when messages are dropped.

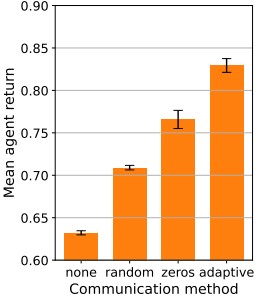

Figure 5: Mean agent return of the pseudo-gradient method with ablations over 5 runs.

## 4.2 TRAFFIC JUNCTION

The traffic junction environment by Sukhbaatar et al. (2016) has been used in several works with slight changes (Das et al., 2019; Singh et al., 2019) to study communication in MARL. We use the version by Singh et al. (2019).

**Environment**  In the traffic junction environment (see Fig. 6), agents control cars that move along predefined routes in a gridworld. Their goal is to reach the end of the route without colliding with other agents. The cars are controlled by a discrete action that is either *gas* or *break*. Action gas advances the car to the next cell in its route, action break leaves the car where it is. If the current number of cars is lower than a predefined number of agents, new cars spawn with probability $p$ at the beginning of each lane. When a car spawns, it is assigned a predefined route that starts at this position. An active

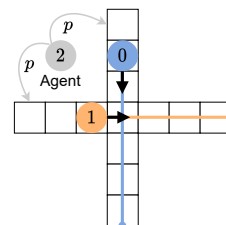

Figure 6: Traffic junction with two agents that control two cars.

car is removed when it reaches the end of the route. The reward for agent $i$ with an active car at step $t$ is defined as $-0.01\tau_i - 10C_i^t$, where $\tau_i$ is the number of steps the car of agent $i$ has been active for and $C_i^t$ is the number of cars that are involved in a collision with agent $i$ at step $t$. Agents that control an active car observe their last action, the route id and information about the agent's position, including the number of cars at this position. They do not observe positions of other agents and need to communicate to prevent collisions. We focus on a two-lane scenario with one route each and up to 5 agents with spawn probability $p = 0.3$. Each episode ends after 20 steps.

**Training**   Our configuration is based on Singh et al. (2019). We use the stochastic action selector and train for 2000 iterations with 128 parallel environments using the Adam optimizer and learning rate 0.001. The core is extended by a gated recurrent unit (Cho et al., 2014). A curriculum increases $p$ from 0.1 to 0.3 between training iterations 250 and 1250. A training run takes around 30 minutes without communication to 1 hour with adaptive communication. Details are in Sec. A.1.

**Results**   The performance in traffic junction is measured in terms of a success rate that is defined as the percentage of episodes without any crashes. Singh et al. (2019) report a success rate of up to $30.2 \pm 0.4$ without communication and $93.0 \pm 3.7$ when agents communicate their continuous-valued hidden state of size 128. We consider message sizes $\Phi \subseteq \{0, 32, 128\}$ and discrete communication with the pseudo-gradient method. We reproduce the results for the baseline without communication with a mean success rate of $30.27 \pm 1.34$. In an unlimited channel, we slightly outperform the baseline with a mean success rate of $97.04 \pm 1.75$. The results with communication are shown in Tab. 2, we focus on the highlighted parts. Communication greatly improves the success rate compared to the baseline without communication, even with messages of size 32 in a limited channel. The success rate for message size 128 under a limited channel declines compared to message size 32, while showing an increasing number of drops. This suggests that reliable communication is critical in this environment. Our approach achieves a higher mean success rate than random message size selection, but the standard deviation of its success rate, throughput and mean message size are very high. We conclude that our approach is unstable in this environment. We think this could be improved with alternative formulations of the message-size value, e.g. by reducing the variance of the target.

Table 2: Results for pseudo-gradient messages. Mean over 5 runs with 2048 evaluation episodes.

| channel size $C$ | 512 | 512 | 512 | 512 | $\infty$ |
|---|---|---|---|---|---|
| message sizes $\Phi$ | $\{32\}$ | $\{128\}$ | $\{0, 32, 128\}$ | $\{0, 32, 128\}$ | $\{128\}$ |
| selection method | fixed | fixed | adaptive | random | fixed |
| success rate | **87.35** $\pm 1.41$ | 64.67 $\pm 1.17$ | **82.78** $\pm \mathbf{9.71}$ | 63.17 $\pm 1.37$ | **97.04** $\pm 1.75$ |
| # of drops | 0.48 $\pm 0.01$ | **1.54** $\pm 0.01$ | 0.64 $\pm 0.49$ | 0.63 $\pm 0.01$ | 0.00 $\pm 0.00$ |
| throughput | 78.63 $\pm 0.47$ | 175.78 $\pm 0.53$ | 93.69 $\pm 40.68$ | 98.69 $\pm 0.71$ | 391.54 $\pm 5.47$ |
| mean msg size | 32.00 $\pm 0.00$ | 128.00 $\pm 0.00$ | 55.97 $\pm 35.86$ | 53.37 $\pm 0.08$ | 128.00 $\pm 0.00$ |

## 5   Conclusion

With this work, we investigate communication over unreliable and limited channels in MARL. We propose an adaptive communication mechanism that allows agents to choose different message sizes depending on their local observations and incoming messages. We introduce the POMNIST environment and show that adaptive communication achieves higher returns than non-adaptive approaches in a limited and unreliable channel. We compare different message types and find that continuous communication performs best, followed by discrete communication with the pseudo-gradient method. The traffic junction environment indicates limitations and open challenges of adaptive communication. Although our approach is better than random message size selection, it is unstable and on average inferior to choosing fixed-size messages in this environment. Future work could investigate the applicability of adaptive communication to different domains and improve its stability, e.g. with alternative formulations of the message-size selection target. Possible extensions with respect to the communication channel include shared medium access control mechanisms and the consideration of more sophisticated channel models. Another possible direction is the interpretation of the message content. A comparison with traditional machine learning methods would also be valuable.

**Reproducibility Statement**  We provide our implementation with the supplementary material together with a README file that describes how to set up and run our code. The included experiment configuration files allow readers to reproduce all experiments that are reported in the main paper and the appendix. Additional details regarding the used models and training procedure are in Sec. A.1.

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

# A APPENDIX

## A.1 TRAINING AND ARCHITECTURE DETAILS

The input of the model consists of the agent's observation and the received messages. Environment-specific observation decoders map observations to a vector of length 128. The observation decoder for POMNIST is a multi-layer convolutional neural network. The first convolutional layer convolves the input with 16 filters of size $3 \times 3$ with stride length of 1 followed by a rectifier nonlinearity. The second layer convolves 32 filters of the same size and stride length. This is followed by a $2 \times 2$ max-pooling layer. The last layer is a fully connected layer with 128 rectifier units, followed by a dropout layer with drop probability 0.5. The observation decoder for traffic junction is a single-layer fully connected network with 128 rectifier units. The message decoder maps all incoming messages to a vector of fixed length. It takes the mean of all messages padded to the same length, including a one-hot encoding of the message's length. First, each message $m$ of size $\varphi$ is padded with zeroes to the maximum message size $\max \Phi$. Then we append a mask that indicates the size of the message as a one-hot encoding of $\varphi$. The output of the message decoder is the mean of these vectors $m \oplus 0^{\max \Phi - \varphi} \oplus \textit{one-hot}(\varphi)$. In practice, we define $\textit{one-hot}(0) := \mathbf{0}$ to indicate empty or dropped messages and mask them during aggregation.

The vectors from the observation and message decoders are concatenated and processed by a core network. The input size of the core network $l_{\text{core}}$ depends on $\Phi$ and is given as $l_{\text{core}} = 128 + \max \Phi + |\Phi|$. In POMNIST, we also append an one-hot encoding of the agent's id, i.e. $l_{\text{core}}$ is extended by $N$. The core module of POMNIST is a fully connected layer with $l_{\text{core}}$ rectifier units and a skip connection. We append a single-layer gated recurrent unit with $l_{\text{core}}$ hidden features for traffic junction. The output of the core network is the input to the following networks.

The message-size selector is a single-layer feedforward network with $|\Phi|$ neurons to predict the message-size values. For exploration, we use exponential $\epsilon$-decay from 1 to 0.01 between iterations 400 to 1200. The message encoder is a multi-headed network with a shared layer of $l_{\text{core}}$ neurons and a separate head for each message size $\varphi \in \Phi$, both followed by tanh activation functions. As mentioned in Sec. 3.2, the action selector is conceptually independent from the rest of the architecture and can be trained using different reinforcement learning algorithms. For POMNIST, we use a single-layer feedforward network to predict the Q-value for each action and an $\epsilon$-greedy action selector with $\epsilon = 0.01$. For traffic junction, we use a stochastic policy that determines the action probabilities with a single-layer feedforward network followed by a softmax. We train this network with REINFORCE and use a separate linear layer to predict the state-value baseline. We encourage exploration by adding an entropy bonus to the loss that is weighted by a factor that decays linearly from 2 to 0.1 over the first 1400 iterations and is 0.1 thereafter.

We jointly train these networks by minimizing $\alpha \mathcal{L}_{\Phi}^{i} + (1 - \alpha) \mathcal{L}_{\pi}^{i}$ over all agents. The mixing coefficient $\alpha$ is set to 0.5 in POMNIST and 0.1 in traffic junction to mitigate the instability. In traffic junction, we use gradient clipping with a maximum 2-norm of 0.1. For both environments, we use the Adam optimizer with learning rate 0.001 and a discount factor of 1.

## A.2 POMNIST ENVIRONMENT CONFIGURATION

**Number of agents** The most influential parameters in the POMNIST environment are the number of splits along the horizontal and vertical axes. They determine the number of agents in the environment. The full view of size $28 \times 28$ is divided according to the number of splits along each axis and we assign one agent to each view. For example, with 0 vertical and 1 horizontal splits, we get two agents with views of size $28 \times 14$ each. The more splits, the higher the number of agents and the smaller the view of each agent. The results for different splits without communication are shown in Tab. 3. As expected, the agents' performance decreases with an increasing number of splits. Note that random guessing corresponds to a mean return of $1 \cdot 0.1 - 1 \cdot 0.9 = -0.8$. Even the agents with $(3, 3)$ splits learn a policy that is much better than random guessing. We decided to continue with $(1, 1)$ splits, as the computational overhead of sending messages will be lower than for 8 or 16 agents and the return is already much lower than the return of the baseline $(0, 0)$. We show that agents can use communication to improve their return in the following.

Table 3: Mean return and standard deviation without communication over 5 runs with different splits.

| (vertical, horizontal) splits | $(0,0)$ | $(0,1)$ | $(1,0)$ | $(1,1)$ | $(1,3)$ | $(3,1)$ | $(3,3)$ |
|---|---|---|---|---|---|---|---|
| number of agents | 1 | 2 | 2 | 4 | 8 | 8 | 16 |
| **mean agent return** | 0.978 | 0.863 | 0.909 | 0.633 | 0.125 | 0.029 | $-0.315$ |
| $\pm$ | 0.001 | 0.001 | 0.001 | 0.003 | 0.012 | 0.004 | 0.011 |

**Message sizes**  Next, we have to choose the message sizes $\Phi$ for our four agents. Higher message sizes allow them to encode and transmit more information, but also require more resources. This can be problematic in limited and unreliable channels. We hypothesize that the benefit of higher message sizes depends on the considered environment and that the resulting performance saturates with an increasing message size.

With continuous messages as our baseline, this performance saturation is already visible for relatively small message sizes in POMNIST. Tab. 4 shows the mean evaluation performance after training with selected message sizes between 0 and 256. While the return increases by more than 0.2 between message sizes 0 and 1, the step-wise improvements diminish with greater message sizes. While the difference between 2 and 4 is still greater than 0.03, doubling the message size from this point on results in improvements smaller than 0.01. Based on these results, we choose message sizes $\Phi \subseteq \{0, 1, 2, 4\}$ for our main experiments. Message size 0 represents not sending any message. Agents can choose this size to keep the channel free if they do not have valuable information to share. Message size 1 allows for a minimum of coordination and message sizes 2 and 4 are a trade-off between a comparably small message size and a high return.

We hypothesize that this saturation effect will also depend on the concrete message type. A continuous messages of size 1 with a 32-bit float can carry significantly more information than a 1-bit discrete message. While this is true in theory, the experiments in our main paper show that agents can perform well even with discrete messages of small sizes.

Table 4: Mean return and standard deviation over 5 runs with fixed-sized continuous messages, i.e. $|\Phi| = 1$, on the test dataset after training for 2000 iterations.

| $\Phi$ | $\{0\}$ | $\{1\}$ | $\{2\}$ | $\{4\}$ | $\{8\}$ | $\{16\}$ | $\{32\}$ | $\{64\}$ | $\{128\}$ | $\{256\}$ |
|---|---|---|---|---|---|---|---|---|---|---|
| **return** | 0.633 | 0.867 | 0.923 | 0.957 | 0.966 | 0.969 | 0.972 | 0.972 | 0.974 | 0.974 |
| $\pm$ | 0.003 | 0.008 | 0.007 | 0.001 | 0.001 | 0.001 | 0.000 | 0.002 | 0.000 | 0.002 |

The optimal choice of message sizes depends on the used hardware and environment-specific factors, e.g. balancing the training overhead and the diminishing improvements in return for higher message sizes. Therefore, this design choice should be made on a case-by-case basis.

Next, we analyze how the return changes depending on the number of received messages with four agents. Tab. 5 shows the mean agent return for message sizes 1, 2 and 3 when an unlimited communication channel only forwards messages from certain agents and blocks all other messages.

Table 5: Mean return and standard deviation over 5 runs with fixed-sized continuous messages and different sender subsets on the test dataset after training for 2000 iterations.

| senders | $\{\}$ | $\{0\}$ | $\{0,1\}$ | $\{0,1,2\}$ | $\{0,1,2,3\}$ |
|---|---|---|---|---|---|
| $\Phi = \{1\}$ | $0.632_{\pm 0.004}$ | $0.774_{\pm 0.003}$ | $0.835_{\pm 0.003}$ | $\mathbf{0.858}_{\pm 0.012}$ | $\mathbf{0.856}_{\pm 0.007}$ |
| $\Phi = \{2\}$ | $0.634_{\pm 0.001}$ | $0.792_{\pm 0.002}$ | $0.877_{\pm 0.008}$ | $0.916_{\pm 0.005}$ | $0.923_{\pm 0.001}$ |
| $\Phi = \{4\}$ | $\mathbf{0.636}_{\pm 0.002}$ | $\mathbf{0.8}_{\pm 0.001}$ | $0.9_{\pm 0.003}$ | $0.946_{\pm 0.002}$ | $0.957_{\pm 0.001}$ |

The gradual increment in the reward for an increasing number of transmitted messages is similar to Tab. 4. There is a big improvement from not transmitting any messages $\{\}$ to transmitting one message $\{0\}$, but the benefit of transmitting more messages diminishes with the number of senders.

Especially for message size 1 ($\Phi = \{1\}$), the difference between transmitting 3 messages from agents $\{0, 1, 2\}$ and 4 messages from all agents $\{0, 1, 2, 3\}$ is neglectable. This suggests that the messages contain redundant information and it might not be necessary or even beneficial to receive messages from all agents at all times. Although the agent's return increases slightly from 3 to 4 sending agents for message sizes 2 and 4, the higher utilization might lead to additional collisions and thus reward detonations in imperfect channels.

### A.3 POSITIVE LISTENING METRIC

In order to quantify the impact of communication on an agent's behavior, we use a *positive listening* metric (Lowe et al., 2019). We define positive listening as the rate at which an agent makes a wrong decision and corrects it after receiving messages. We compute this metric in POMNIST by exploiting the two-step execution explained in Sec. 4.1. The messages are considered at the second step of the decision making, while in the first step the agent takes only its observation into account. According to this, we measure the effect of messages by comparing the chosen action in the first and second step. It should be noted that the models used for the computation of this metric must satisfy the condition $0 \in \Phi$ and have no internal state, i.e. the agents are trained to make the decision without any messages.

### A.4 POSITIVE SIGNALING METRIC

To quantify positive signaling in discrete action and message spaces, Lowe et al. suggest to use the mutual information between selected actions and sent messages (Lowe et al., 2019):

$$I(\mathcal{A}; \mathcal{M}) \coloneqq \sum_{a \in \mathcal{A}, \, m \in \mathcal{M}} p(a, m) \log \frac{p(a, m)}{p(a)p(m)} \tag{3}$$

They treat the action space $\mathcal{A}$ and discrete message space $\mathcal{M}$ as random variables that take on values $a \in \mathcal{A}$ and $m \in \mathcal{M}$ with probabilities $p(a)$ and $p(m)$ respectively. In practice, these probabilities are approximated empirically by normalizing the action and message frequencies.

In adaptive communication, agents can choose from different message sizes $\varphi \in \Phi$. Messages for each size are chosen independently and the entropy of the corresponding message distributions can differ. Additionally, messages of certain sizes could be solely sent in conjunction with certain actions. As the mutual information is non-negative and bound by the minimum entropy of actions and messages, i.e. $0 \leq I(\mathcal{A}; \mathcal{M}) \leq \min\{H(\mathcal{A}), H(\mathcal{M})\}$, we can normalize the mutual information across different message sizes.

Based on these considerations, we express positive signaling jointly for all messages sizes as

$$PS \coloneqq \sum_{\varphi \in \Phi \setminus \{0\}} \frac{p(\varphi)}{1 - p(0)} \frac{I(\mathcal{A}_\varphi; \mathcal{M}^\varphi)}{\min\{H(\mathcal{A}_\varphi), H(\mathcal{M}^\varphi)\}} \tag{4}$$

where $p(\varphi)$ is the probability of choosing a message of size $\varphi \in \Phi$ and $\mathcal{M}^\varphi$ is the set of messages with this size. The random variable $\mathcal{A}_\varphi$ conditions actions on message size $\varphi$. If $0 \notin \Phi$, we define $p(0) \coloneqq 0$. Note that empty messages of size $\{0\}$ are excluded from the metric, i.e. an agent that communicates only when taking a specific action in a specific state and stays silent otherwise will have a positive signaling value of 1 instead of 0.

Note that our positive signaling metric is defined for discrete messages. As continuous messages can take any value in range $[-1, 1]^\varphi$, they are often unique. This would lead to mutual information and positive signaling values of 1 while not allowing for any conclusions regarding the usage of the message space with respect to the taken actions. Positive signaling metrics for continuous messages could e.g. be based on clustering methods.

### A.5 COMMUNICATION CHANNEL

In this section, we provide further details regarding the communication channel introduced in Sec. 3.4.

First, we describe an example to illustrate why the expected throughput (see Sec. 4.1) is usually lower than the channel size with the given model. Recall that the starting position for a message of size $\varphi$ is chosen randomly from $\{0,\ \varphi,\ 2\varphi,\ \ldots,\ \lfloor \frac{C-\varphi}{\varphi} \rfloor \varphi\}$ with channel size $C$ and that the message occupies $\varphi$ slots. We call this channel model *stochastic channel with spacing*. Consider 4 agents that try to send messages of size 4 simultaneously in a channel of size $C = 8$. The message of an agent is then placed into slots $\{0, 1, 2, 3\}$ or $\{4, 5, 6, 7\}$, each with probability $0.5$. It is transmitted correctly without collisions iff the messages of all remaining agents are placed into the other slots. With four agents, these are $2 \cdot 4 = 8$ events with probability $0.5^4$. The probability of successfully transmitting any message is therefore $8 \cdot 0.5^4 = 0.5$. As all messages have size 4, the throughput is $0.5 \cdot 4 = 2$.

When 4 agents send messages with random sizes $\varphi \in \Phi = \{0, 1, 2, 4\}$ over a channel of size 8, we empirically get a throughput of 2.297 over 1 million steps. The drop probabilities for different message sizes are shown in Tab. 6. The table also includes the drop probabilities for the *stochastic channel*, where a message of size $\varphi$ is randomly placed into $\{0, 1, 2, \ldots, C - \varphi\}$ and occupies $\varphi$ slots. The results show that this channel access scheme results in a lower throughput of 1.579 and higher drop probabilities for higher message sizes.

Table 6: Message drop probabilities for individual message sizes and throughput using a random message size selection in different channel models.

| Channel model | Drop probabilities | | | | Throughput |
|---|---|---|---|---|---|
| | $\varphi = 0$ | $\varphi = 1$ | $\varphi = 2$ | $\varphi = 4$ | |
| stochastic with spacing | 0 | 0.524 | **0.578** | **0.756** | **2.297** slots |
| stochastic | 0 | **0.512** | 0.682 | 0.886 | 1.579 slots |

Based on these results, we decided to use the stochastic channel with spacing for our main experiments. Further experiments, e.g. comparing the learned behavior for different channel models, are left for future work. An active placement of messages in the channel could also be investigated, e.g. agents could listen on the channel and decide when to send messages to avoid collisions.

## A.6 MESSAGE SIZES

In this section we take a closer look at the mean message sizes at the experiments with 4 agents and message sizes $\Phi = \{0, 1, 2, 4\}$. Fig. 7 shows the mean message size during training. We observe that, for small channel sizes 4 and 8, the agents do not exploit the channel and decrease their message sizes to avoid collisions. However, with higher channel sizes ($C \geq 16$), the agents first try to exploit the resources and the mean message size increases from the random selection. As expected, they learn to decrease their message size to avoid collisions later in the training. Therefore, the mean message size decreases and converges to a message size that is higher than the random selection. The message sizes after training for 2000 iterations are reported on Tab.1 on the main paper for different channel sizes.

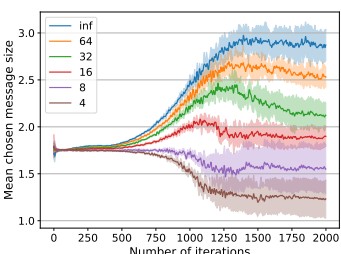

Figure 7: Mean chosen message size during training with different channel sizes. Each line shows the mean over 5 runs, the transparent area represents the standard deviation.

