# OpenReview forum: "Learning to Cooperate and Communicate Over Imperfect Channels"
_ICLR.cc/2023/Conference — Submitted to ICLR 2023_

### Official Review · Reviewer_rgRs · 2022-10-20

**Confidence:** 4
**Correctness:** 3
**Technical Novelty And Significance:** 2
**Empirical Novelty And Significance:** 2
**Recommendation:** 3

**Clarity, Quality, Novelty And Reproducibility:**

The paper is sufficiently well-written. Unfortunately, there are some details of the work that are not completely clear - see above.

In general, the originality of the work is somehow limited given its generalizability and the novelty of the architecture itself.

**Strength And Weaknesses:**

This is an interesting paper about a problem of general applicability. The main concern is that the problem itself is evaluated in a very specific environment. The main novelty of the work resides probably in framing the general problem of learning the type of messages exchanged by the agents in presence of noise. The actual solution per se is rather standard (end-to-end training with local encoders that generate messages that are exchanged among a group of agents in order to execute a task).
More specifically, the reviewer wonders if the actual task used for the evaluation might have a strong impact on the results. In fact, at the end, the local "encoding" might be seen as a problem of classification. Each agent "encodes" the most probable "digits" that are present in the image and then this is used for the overall classification. There might be different types of encoding given different available sizes. From the results presented by the authors, there is a minimal amount of information that is necessary to transmit. The problem of traffic in this case can be framed as communication over a noisy channel, but then, again, a "trivial" solution might be to have a sufficient amount of information sent. It would be interesting to see what happens when just one or two agents transmit their messages, just to check if the results are independent from the fact that all (or most of) the agents are involved. Having said that, I think that the authors present an interesting set up for testing their model. The question is if it is possible to have sufficient evidence using only that environment in terms of validation. Probably different tasks that are not related to classification are necessary for a full-evaluation of this work. This is quite important, since the solution is evaluated only experimentally.

There is also a question about the message that is sent - the actual selection of the size of the message should be clarified in the paper.

There is also a question about which actual component has the strongest influence/impact on the obtained results. I understand that this is kind of lost in an end-to-end training, but I think it is a really important aspect. It seems that the size is important, but probably mostly importantly the encoding efficiency (given the probability of transmission error ) is also a key factor.

In general, the paper only briefly discusses the limitations of the proposed solution. A major limitation of the work appears to be the fact that the authors consider only one rather specific task (distributed image classification). It is quite difficult to generalize from the experimental results they present, even if they are valuable.

Other open points about this paper are the following:

- The reviewer wonders if the results we observe might be considered as strictly related to the problem/environment under consideration. How can you say that the results reported in the paper might be generalized to other environments?

- What is the real impact of the encoding, etc. when you have redundancy in terms of information? It seems to me that this work applies to cases where every agent is essentially performing the same action. Is it possible to extend this work to situations in which the different agents are involved in different tasks and communication is needed for coordination?

- An interesting aspect to investigate is the fact that you have overlapping "states" (different views of the image). Is it possible to say that this is not a factor contributing to the results presented in the paper or is it?







**Summary Of The Paper:**

The paper discusses a solution for communication in multi-agent learning in terms of message size (and implicitly when to transmit). The paper is evaluated by means of an environment developed by the authors, which is based on a classification task of the MNIST dataset based on partial views of the images themselves. Each agent "decides" which information is sent to the others in order to execute the classification task end-to-end.

**Summary Of The Review:**

Overall, the reviewer struggles to see a sufficient contribution in this paper. The reviewer has concerns about the motivation, generalizabilty and design of the architecture. For this reason, the reviewer is not in a position to recommend this work for publication.

---

> ### Author Response · Authors · 2022-11-18
> **Response to Reviewer rgRs**
>
> Thank you for your review. We have evaluated our approach in POMNIST and the traffic junction environment (see section 4.2). Our approach is not specifically designed for nor restricted to distributed image classification. It is applicable to any MARL problem with communication channels as introduced in section 3.1. This includes problems where different agents perform different tasks, e.g. cars going in different routes in the traffic junction environment. However, our experiments show that the combination of our method and the underlying learning algorithm is unstable in this environment. We therefore focused on POMNIST for this paper.
>
> The observations in POMNIST are not overlapping, each digit is split into non-overlapping views for each agent. In general, observations in MARL can be overlapping (e.g. in a multi-agent particle environment). We think that such scenarios would be interesting, agents could choose smaller message sizes the more their observations overlap. This could also be included as a new setting in POMNIST.
>
> Our experiments show that transmitting very little information (e.g. a single bit) already improves the performance in POMNIST. However, this information is not “sufficient” in the sense that the agents can still achieve significantly higher returns with higher message sizes. If the communication channel allows for a transmission of bigger messages, then of course the “trivial” solution would be to always choose the greatest message size or one for which the return saturates (if we ignore diminishing returns due to a higher number of parameters). However, this does not work if the channels have insufficient capacity. Our work shows that selecting a fixed message size for all agents can be suboptimal (see Fig. 3b) when agents have to communicate over an unreliable and limited channel.
>
> We provide further details regarding the selected message sizes and experimental results for different sets of sending agents in section A.2 in the appendix.

---

> > ### Comment · Reviewer_rgRs · 2022-12-04
> > **Thanks for the response**
> >
> > Many thanks for your comments. I found your explanation about the contents of the message insightful, but your reply does not fully address my concerns, especially in terms of generalizability.

---

### Official Review · Reviewer_JWRp · 2022-10-25

**Confidence:** 3
**Clarity, Quality, Novelty And Reproducibility:** The paper is clear, of reasonable qua…
**Correctness:** 3
**Technical Novelty And Significance:** 3
**Empirical Novelty And Significance:** 2
**Recommendation:** 5

**Strength And Weaknesses:**

Strengths:
- Multiagent communication and cooperation are important questions that are topics of broad interest to the community.
- The problem of limited and noisy channels is an important one in practice. (At least noisy channels is for humans.)
- The proposed approach extends prior work toward an interesting direction.
- The POMNIST problem is fun.
- It is a positive that the traffic junction problem reveals limitations of the approach.

Weaknesses:
- The problem itself, while obviously necessary, is not so interesting. The novelty contribution seems relatively modest.
-  There is no theoretical contribution.

**Summary Of The Paper:**

The paper presents a reinforcement learning based approach to multi-agent communication over noisy or limited channels. Agents therefore adapt to share messages of varying lengths and content depending on the channel characteristics and the local environment. The approach is demonstrated on a partially observable version of MNIST (POMNIST) in which each agent is allowed access to a mutually exclusive and jointly exhaustive subset of the image, and on a traffic junction problem. The results illustrate the ability adapt to different characteristics of channels and successfully communicate to achieve shared goals.

**Summary Of The Review:**

The paper addresses the problem of communication about partially observed worlds over unknown channels. The proposed approach uses reinforcement learning to dynamically adapt to the features of the world and the channel so that agents may cooperate toward shared goals. While the problem and approach seem to be novel, the contribution appears to be modest.

---

> ### Author Response · Authors · 2022-11-18
> **Response to Reviewer JWRp**
>
> Thank you for your review! We are happy to hear that you consider the overall direction of our work and the POMNIST problem interesting. We will aim to improve the theoretical foundation and understanding of our approach in the next iteration of this work.

---

### Official Review · Reviewer_hSti · 2022-10-30

**Confidence:** 3
**Correctness:** 3
**Technical Novelty And Significance:** 2
**Empirical Novelty And Significance:** 2
**Recommendation:** 5

**Clarity, Quality, Novelty And Reproducibility:**

The paper is mainly clear and the quality of the writing is satisfactory. The contribution of the paper sounds limited. The authors provide code, so the paper should be reproducible.

**Strength And Weaknesses:**

**Strengths**
- The topic of communication-efficient cooperative learning is a timely topic.
- The focus of this paper on considering the imperfect communication channels is new in the context of MARL.
-  The paper is well-written and easy to read.
- The experiments are given using two datasets.

**Weaknesses**
- While the imperfect channel issue makes sense and is important, the paper does not provide any theoretical understanding and analysis of the proposed cooperative policies.
- Hence, this paper should be evaluated as an empirical study. Then, focusing just on experiments, it seems that the paper fails to compare with prior cooperative policies. In the current presentation, it is unclear whether the algorithms are from other papers or are different versions of the algorithms proposed in this paper.
- In addition, the authors can provide some insights into the observations from the experiments. For example, for Fig. 3a, the authors just state which algorithm is the best, and there are no further justifications for the reasoning of these results.


**Summary Of The Paper:**

This paper proposes a communication policy for multi-agent RL when the communication channel is imperfect. The idea is to adapt the size of communication data based on observations from the communication channel. Then, the performance of the proposed policies was evaluated using two datasets. The results show that the proposed algorithms outperform the alternatives.

**Summary Of The Review:**

Overall, this is a timely topic. However, the contribution of this paper sounds limited. On the theory side, there is no theoretical analysis of the proposed algorithms. On the experimental side, there could be more comparison with the state-of-the-art and also more insights on the results.

---

> ### Author Response · Authors · 2022-11-18
> **Response to Reviewer hSti**
>
> Thank you for your feedback! It would be interesting to combine our approach with prior cooperative policies that include message exchange, but we find that a direct comparison would be unfair as these methods typically assume a perfect communication channel. They would underperform if they were applied in limited channels without modifications (e.g. see the low success rate with fixed message size 128 in Tab. 2). Because of this, our idea was to first compare different message encoders and then to evaluate our method under different conditions using the empirically best encoder.
>
> Thank you for your remarks regarding the discussion of our results, we will provide further justifications and reasoning in future iterations of this work.
>
> The learning algorithms behind the action selection are from related work and not part of our contribution (see “action selector” in Sec. 3.2). Moving forward, we aim to improve the theoretical understanding of our method and to clearly separate our contribution from the underlying learning algorithm(s) and message types.

---

### Decision · Program_Chairs · 2023-01-20

**Decision:**

Reject

**Justification For Why Not Higher Score:**

All the weaknesses mentioned above.

**Justification For Why Not Lower Score:**

N/A

**Metareview: Summary, Strengths And Weaknesses:**

This paper proposes to study the problem of learning to adapt communication over imperfect channels as a task for multi-agent reinforcement learning. Concretely, a "partially observed MNIST (POMNIST)" task is proposed and methods are studied empirically in this setting.

Some weaknesses include:
* No theory is provided to motivate the proposed task/problem setting or to support the empirical observations
* As an empirical-only work, the paper could be strengthened by comparing with previous work, and including additional experiments to better illustrate insights of the current experiments (not just focusing on better performance/numbers). The empirical results could also be strengthened by expanding to other environments or otherwise creating more variety to provide a more convincing argument for generality of the proposed approach.

**Summary Of Ac-Reviewer Meeting:**

n/a